# Resilience and Equity in a Time of Crises: Investing in Public Urban Greenspace Is Now More Essential Than Ever in the US and Beyond

**DOI:** 10.3390/ijerph18168420

**Published:** 2021-08-09

**Authors:** Jean C. Bikomeye, Sima Namin, Chima Anyanwu, Caitlin S. Rublee, Jamie Ferschinger, Ken Leinbach, Patricia Lindquist, August Hoppe, Lawrence Hoffman, Justin Hegarty, Dwayne Sperber, Kirsten M. M. Beyer

**Affiliations:** 1Institute for Health & Equity, Medical College of Wisconsin, 8701 Watertown Plank Rd., Milwaukee, WI 53226, USA; jbikomeye@mcw.edu (J.C.B.); snamin@mcw.edu (S.N.); canyanwu@mcw.edu (C.A.); 2Department of Emergency Medicine, Medical College of Wisconsin, 8701 Watertown Plank Rd., Milwaukee, WI 53226, USA; crublee@mcw.edu; 3Sixteenth Street Community Health Centers, Environmental Health & Community Wellness, 1337 S Cesar Chavez Drive, Milwaukee, WI 53204, USA; jamie.ferschinger@sschc.org; 4The Urban Ecology Center, 1500 E. Park Place, Milwaukee, WI 53211, USA; kleinbach@urbanecologycenter.org; 5Wisconsin Department of Natural Resources, Division of Forestry, 101 S. Webster Street, P.O. Box 7921, Madison, WI 53707, USA; patricia.lindquist@wisconsin.gov; 6The Urban Wood Lab, Hoppe Tree Service, 1813 S. 73rd Street, West Allis, WI 53214, USA; info@hoppetreeservice.com; 7Department of GIS, Groundwork Milwaukee, 227 West Pleasant Street, Milwaukee, WI 53212, USA; lawrence@groundworkmke.org; 8Reflo—Sustainable Water Solutions, 1100 S 5th Street, Milwaukee, WI 53204, USA; justin.hegarty@refloh2o.com; 9Wudeward Urban Forest Products, N11W31868 Phyllis Parkway, Delafield, WI 53018, USA; dwayne@wudeward.com

**Keywords:** greenspace, urban neighborhoods, health equity, COVID-19, climate change, cancer, cardiovascular diseases (CVD), structural racism, resilience, health disparities

## Abstract

The intersecting negative effects of structural racism, COVID-19, climate change, and chronic diseases disproportionately affect racial and ethnic minorities in the US and around the world. Urban populations of color are concentrated in historically redlined, segregated, disinvested, and marginalized neighborhoods with inadequate quality housing and limited access to resources, including quality greenspaces designed to support natural ecosystems and healthy outdoor activities while mitigating urban environmental challenges such as air pollution, heat island effects, combined sewer overflows and poor water quality. Disinvested urban environments thus contribute to health inequity via physical and social environmental exposures, resulting in disparities across numerous health outcomes, including COVID-19 and chronic diseases such as cancer and cardiovascular diseases (CVD). In this paper, we build off an existing conceptual framework and propose another conceptual framework for the role of greenspace in contributing to resilience and health equity in the US and beyond. We argue that strategic investments in public greenspaces in urban neighborhoods impacted by long term economic disinvestment are critically needed to adapt and build resilience in communities of color, with urgency due to immediate health threats of climate change, COVID-19, and endemic disparities in chronic diseases. We suggest that equity-focused investments in public urban greenspaces are needed to reduce social inequalities, expand economic opportunities with diversity in workforce initiatives, build resilient urban ecosystems, and improve health equity. We recommend key strategies and considerations to guide this investment, drawing upon a robust compilation of scientific literature along with decades of community-based work, using strategic partnerships from multiple efforts in Milwaukee Wisconsin as examples of success.

## 1. Introduction

The world is currently facing several intertwined and urgent global health crises—climate change [1,2], COVID-19 [3], and structural racism [4]—as well as the endemic challenges of chronic conditions such as cancer and cardiovascular diseases (CVD) that are persistent leading causes of death. The confluence of these crises, long in the making, has brought health inequities into sharp focus, particularly in the US, where a highly polarized political landscape has magnified these challenges and policymaking for each crisis has become intertwined with the battle for political power. Importantly, these four concurrent crises constitute a web of social and environmental risk factors that negatively impact health, disproportionately impacting people of color. These challenges occur in addition to decades of health inequities attributable to a range of factors, including racial residential segregation, environmental pollution, economic disinvestment, poor-quality schools and housing, socio-economic deprivation, food insecurity and inadequate access to health care [5], all negatively and disproportionately affecting communities of color. Further, the urgency of these crises challenges populations to cope and muster resilience in the context of neighborhoods and systems with inadequate safety nets and among populations with little accumulated wealth to fall back on, particularly in the US, given the intimate connections between economic investment, home ownership and wealth and linkages between employment and access to health care [6,7]. In this paper, we discuss the intersecting problems of structural racism, climate change, COVID-19 and chronic diseases and argue that strategic investments in public urban greenspace can, and should be part of the solution, offering resiliency for communities long subject to inequitable economic investment and offering opportunities for primary prevention of public health disparities. Here, we define public urban greenspace broadly, to include urban trees, forests, parks, natural areas, community gardens, and green infrastructure that are accessible to communities in urban areas, while recognizing the additional benefits of private greenspaces that offer similar benefits. We offer strategies and considerations to guide this investment, drawing upon decades of successful community-based work and partnerships and a robust scientific literature.

## 2. Four Intersecting Crises: Structural Racism, COVID-19, Climate Change, and Chronic Diseases

### 2.1. Structural Racism and Health Outcomes

Significant racial and ethnic inequities and health disparities are pervasive and persist today because of longstanding structural racism [8,9,10]. Structural racism comprises the historical and contemporary ways in which society maintains discriminatory practices that mutually reinforce inequitable systems across a variety of sectors [8,10,11], including housing, education, employment, income, criminal justice, and healthcare [10,12,13]. For example, zoning policies, racially restrictive covenants, and segregation have created a pathway to socio-economic disinvestment and neighborhood deprivation [14]. Neighborhood deprivation is linked to limited access to employment, inadequate quality education, housing instability, reduced access to quality healthcare, and other social risk factors which intersect as fundamental causes of disease and poor health outcomes [11].

The COVID-19 pandemic has provided a painfully clear illustration of how structural racism, socioeconomic vulnerability, and health outcomes are intertwined. From the early months of the pandemic, it was apparent that rates of infection and mortality were higher in communities of color [15]. Discussions around the role that the federal government should have played in alleviating the burden of disease in vulnerable communities ensued [16]. The murder of George Floyd on 25 May 2020 brought structural racism again into focus. Though many videos of police violence against people of color had surfaced prior, the nature of the crime (9 min and 29 s of footage showing the slow death of Mr. Floyd while onlookers sought to intervene), the existing frustration of the pandemic, the increasingly high profile of the Black Lives Matter movement, and a national sense of political division resulted in widespread and extensive protests and the words “structural racism” took hold in the public discourse. This new, broad recognition of structural racism is an essential building block toward change, but it is critical also to recall the historical roots of impoverishment and vulnerability in communities of color to move forward.

#### 2.1.1. A Brief History of Structural Racism in American Cities

The story of structural racism in urban housing in the US is long and unresolved. This story is dependent on an even older story of the social construction of race and the use of racial categories to systematically disadvantage specific population groups in myriad ways [17]. It must also be noted that the urban focus of this review omits an in-depth consideration of structural racism as a factor in the segregation of Native American Indian populations on rural reservation land and the many adverse consequences of this for native health. While parallels exist for Black individuals, Indigenous peoples, and people of color (BIPOC), the focus of this work is on Black urban populations.

The patterns of residential racial segregation that we observe in US cities today can be traced back to the institution of American slavery and to the post-civil war reconstruction era (1865–1877), during which slavery was outlawed [18], but social reconstruction failed. The labor system structures shared similar characteristics with the pre-civil war era, and existence of racist codes and policies resulted in increased inequality between Black/African Americans and European Americans [19,20]. As Du Bois (1999) put it [21], the reconstruction era saw the “self-assertion of the white race against the impudent ambition of the degraded blacks’’ [22], which resulted in disfranchisement of Black people. This era helped to create “unequal forms of social existence” [23] activated through employment opportunities, caste education and the rise of racist groups in addition to government policies that were all in line with racial capitalism [23]. The reconstruction era created a system of segregation that then grew into the post-Reconstruction era known as “Jim Crow” that further reinforced legal segregation through the US Supreme Court [24] and isolated people of color in overcrowded and poor housing away from white neighborhoods [25].

Escalating racial violence under Jim Crow laws, along with constraints in job opportunities and environmental disasters, motivated what is known as the Great Migration—the migration of Black Americans from southern to northern states from 1916 to 1970 [26]. The Great Migration has been analyzed as a contributor to the racial gap in housing and health that exists to this day. It changed the racial composition of the destination cities in the North [27] and reduced public spending and tax revenues in northern cities, due to reduced property values [28]. Research shows that policy change responses to the new wave of immigrants from the South explains 27% of today’s regional racial upward mobility gap [21]. From the beginning of this migration wave, some anticipated both upward and downward health outcomes for the Black immigrants to the North [29,30,31], and some reports echoed concerns about the adverse health effects of the urban areas occupied by these new immigrants [32].

In the North, people of color faced racialized housing policies in the New Deal era when opportunities for home ownership were systematically limited through redlining. In the 1930s, the Home Owners’ Loan Corporation (HOLC) was created as part of Franklin Roosevelt’s New Deal to address the growing problem of home foreclosures during the Great Depression [33]. In this process, urban neighborhoods were graded on their desirability for investment, and the presence of racial minorities was considered heavily in the auditing process. In the redlining audit process, evaluators pointed to specific populations in describing their grades, such as populations of “undesirables—aliens and negros” [34,35]. On their rating sheets, they checked off a list of nuisance factors, including odors, noises, fire hazards, and *“infiltrations of lower grade population or different racial groups”* [34]. Consequently, research has shown that despite regulations that ended the legal practice of discrimination, HOLC appraised cities have become more segregated than their ungraded counterparts [36] and HOLC appraisals impacted homeownership, property values, housing supply, population, and housing finance outcomes in the long term [37,38,39,40] including current inequalities in access to quality greenspace, social capital as well as the related adverse health outcomes [41]. Formerly redlined areas also have an increased risk for climate change related hazards. For example, formerly redlined areas have USD 107 billion worth of homes facing high flood risk—25% more than non-redlined areas [42,43].

These discriminatory practices led not only to racial segregation patterns, but also to disparities in home ownership. Home ownership has been said by American Presidents on both sides of the aisle to be that which “lies at the heart of the American Dream” [44,45]. Owning a home brings housing stability and is also a key approach to accumulating wealth and achieving financial stability for many Americans. Based on the US Census Bureau 2019 quarterly report, today only 40.6% of Black individuals and 46.6% of Hispanic individuals own their home compared to 73.1% of White individuals [46]. Housing is the largest source of wealth for American households and therefore discrimination in acquiring and financing housing has played a significant role in widening the racial wealth gap [47,48,49].

Today, Black families own a dime for every dollar owned by White families [50]. Approximately 19% of Black households have zero net worth and almost 50% of Black households cannot afford to buy a home [51]. “The median White family has respectively 41 times and 22 times more wealth than the median Black and Latino families” [52]. Moreover, a recent study by the Brookings Institute shows that across metropolitan areas in the US, in majority Black neighborhoods over 50% of home values are at roughly half the price of homes in neighborhoods with no Black population [53]. This study further indicates that even after controlling for home and neighborhood characteristics, and residents’ purchasing power, homes in majority Black neighborhoods are devalued by 19 to 22 percent, compared to homes in non-majority Black neighborhoods [53].

#### 2.1.2. Structural and Environmental Racism

Environmental racism in the US is closely intertwined with structural housing discrimination. Decades of segregation reinforcement through discriminatory housing practices has resulted in distinct patterns of racial segregation across the US that are persistent and contribute to an ongoing crisis of environmental injustice and environmental racism [54,55]. Environmental racism entered the public policy discourse after the Love Canal tragedy in 1978 and polychlorinated biphenyl (PCB) landfills in Warren County, North Carolina in 1982, which led to debates over Environmental Justice (EJ) rights and to studies on the distribution of environmental risks [56]. The United Church of Christ sponsored study in 1986, *Toxic Wastes and Race in the USA: A National Report on the Racial and Socioeconomic Characteristics of Communities with Hazardous Waste Sites,* found that race was the most significant variable in siting hazardous waste facilities and coined the term “environmental racism” [57].

There is no shortage of evidence of environmental racism. Historically, the focus of EJ work has focused on the disproportionate siting of hazardous wastes treatment, storage, and disposal facilities, dumps, polluting industries, byproducts of municipal landfills, and incinerators in community of color’s neighborhoods [58]. Another example is the *cancer alley*, an 85-mile stretch of the Mississippi river that connects Baton Rouge to New Orleans in Louisiana and is home to over 150 petrochemical plants and refineries that emit tons of airborne pollutants [59]. The communities closest to those plants are mainly people of color. More recently, the struggle against the Dakota Access Pipeline in South Dakota, in the name of water quality (“Water is Life”), illustrates the ongoing struggle for environmental rights among native populations in the US [60].

Many public and community health issues are heavily dependent on place [61] and segregation policies have fundamentally contributed to differences in residential environmental exposures by race and ethnicity, including unstable and poor-quality housing, hazardous wastes, industrial emissions, and poor outdoor and indoor air quality [62]. Institutional neglect in poor communities of color has led to increased exposures to environmental toxins through state-sanctioned placement of hazardous wastes and disproportionate siting of coal fired power plants and petrochemical industries in minority neighborhoods [55,63,64]. Moreover, disinvestment in poor minority neighborhoods confers a host of social environmental risk factors such as poverty and crime, which have increased chronic exposures to discrimination, criminal victimization and violence through increased police brutality and incarceration of young adult males of color [9,65,66]. The resulting social conditions reinforce patterns of discrimination in the housing market leading to evictions, unstable housing, unemployment, food insecurity, and unhealthy social behavior as negative coping mechanisms [9,10,67,68]. Discrimination fosters stress and adversely impacts health [69]. All these factors impact health and disease in several ways, including by increasing the risk of CVD, respiratory diseases, cancer and other chronic conditions [63,70,71,72,73].

The range of environmental exposures to which urban populations of color are subjected unsurprisingly produces clear health inequity gradients in downstream health outcomes across numerous health indicators [74,75]. For example, non-Hispanic Blacks have a disproportionate rate of psychological distress, infant mortality and age adjusted mortality rates for chronic diseases such as cancer, diabetes, and CVD, among others [74]. Similarly, indigenous, or native American populations continue to disproportionately suffer from health inequities [76], bearing a higher burden of chronic diseases including obesity, diabetes, chronic liver disease, and CVD attributable to increased risk factors such as geographic isolation, inadequate access to healthcare, poverty, unemployment, food insecurity, low incomes, and inadequate sewage disposal [77].

### 2.2. Structural Racism and COVID-19

Structural racism has contributed to COVID-19 disparities. The situation facing essential workers during the pandemic has illustrated how structural racism is perpetuated in occupational settings through overrepresentation of people of color in low-paying jobs [78,79,80]. High representation in those lower paying jobs has exacerbated social vulnerabilities among people of color in a time of health and economic crisis [81]. Individuals in low-paying “essential” jobs such as grocery store attendants, clinical assistants and meat packing plant workers have been shown to be at increased the risk of COVID-19 infection through lack of ability to physically distance and higher likelihood of encountering an infected person [82]. Consequently, COVID-19 disproportionately affected communities of color, including Black, Indigenous, and Latinx populations [78,79,83,84,85]. For example, in a study that assessed the differential occupational risk of COVID-19 by race, people of color were found to be more than twice as likely to be employed in animal slaughtering and processing industry where notable outbreaks of COVID-19 have been reported, presenting a disproportionate risk for COVID-19 infection [82].

Similarly, in the agriculture sector, inequities in wealth accumulation, one of the legacies of systemic racism, also led to differential occupational distribution and land ownership [86]. Racist practices systematically discriminated against Black and Indigenous farmers, leading to foreclosures, and forcing them into a cycle of debt and poverty [86,87] and limited their ability for land ownership and food sovereignty. For instance, White people own 98% and operate 94% of all farmland [86]. White individuals also generate 98% of all farm ownership related income and 97% of all farm owner-operatorship income [86]. On the other hand, 85% of farm labor is undertaken by people of color, who own less land, are more likely to be tenants with a very tiny portion of farm income [86] while carrying the excess farm labor related risks including higher infection rates for the COVID-19 [88,89,90]. Relative excess mortality during the COVID-19 pandemic was highest in food/agriculture workers, which experienced a 39% increase [88]. These estimates might have been lower than the actual rates because a sizable percentage of farm workers do not have access to basic labor protections such as paid sick leave or access to healthcare [91], which might have limited access to COVID-19 testing.

In addition to differential occupational COVID-19 exposure, neighborhood disadvantage also contributed to increased risk of COVID-19 among communities of color [92]. The sizable number of people of color living in disadvantaged neighborhoods with limited access to social resources such as outdoor greenspaces to enable social connection while physically distancing has contributed to their increased vulnerability to the COVID-19 pandemic [92,93]. Longstanding inequities have contributed to poverty and poor-quality housing situations, characterized by overcrowding and poor indoor air quality, which increased COVID-19 vulnerabilities and infection [94]. Even more, lack of access to pharmacies or “pharmacy deserts” plague many of these urban neighborhoods despite reliance on medications and increased exposures [95]. Recent studies have also shown that environmental exposures such as air pollution support the spread and lethality of COVID-19 [96,97,98,99], a potential partial explanation for increased risk of COVID-19 morbidity and mortality among Black Americans in segregated communities. Air pollution and exposure to air pollutants such as particulate matter (PM_2.5_) disproportionately and systemically affect people of color [100]. Those air pollutants have also been associated with respiratory diseases such as asthma, lung cancer, and chronic obstructive pulmonary disease (COPD) [101] with a disproportionate risk for people of color living near hazardous and toxic waste locations [57,102,103], and other major sources of pollution [104,105]. Studies have found associations between residential proximity to waste sites, coal-fired power plants and hospitalizations for respiratory diseases [101,106,107]. Air pollution can also worsen existing respiratory conditions including asthma and COPD [108].

Similar pattern of association between air pollution and COVID-19 were observed. For example, a study in California found significant association between COVID-19 total cases and death with criteria air pollutants such as particulate matter (PM_2.5_ and PM_10_), nitrogen oxide (NO_2_) and sulfur dioxide (SO_2_) [98]. Another study also found that proximity to hazardous wastes, storage and disposal facility are associated with COVID-19 prevalence and fatalities [97]. Residential neighborhood risk factors with a disproportionate exposure to hazardous and toxic wastes as well as air pollution factors might have intersected in predicting negative outcomes in COVID-19 disease among communities of color.

Further, the lack of financial resources and reduced likelihood of having health insurance limit access to healthcare services among low-income communities of color, as they avoid using an excessively expensive healthcare system [10]. Studies show that many Black and Latinx families are more cost burdened and devote over 50 percent of their family income to pay rent and utilities [68]. In Milwaukee, Wisconsin, the energy burden for Black and Latinx households was more than double that of nearby white neighborhoods at 5.0%, 5.3% and 2.1%, respectively; and this has been linked with redlining [109]. These neighborhoods have been more difficult to reach with information on energy efficient programs and initiatives to reduce costs [110]. Thus, a substantial proportion of income is spent on housing costs, leaving these families with little to spend on other basic human needs including food and health care.

Racial residential segregation [111] and occupational segregation have both been identified as predisposing factors in increasing vulnerability of people of color to COVID-19 infection and its adverse health outcomes. In addition to increased risk of COVID-19 infections, a disproportionate prevalence of comorbid conditions among people of color and other minority communities [112] increases their health care needs, health care services utilization as well as associated costs [113], which cumulatively put extra pressure on their limited income and fortify the need to maintain employment, given the linkage between employment and health insurance in the US [114].

The intersection of multiple social risk factors including lower home ownership rates, co-housing of many generations, large families, working essential jobs, limited access to labor protections, and limited access to COVID-19 testing contributed to higher COVID-19 exposure and prevalence rate as well as the associated adverse health outcomes among people of color [81,115,116] such as increased risk of a positive test for COVID-19, increased likelihood of COVID-19 related hospitalization as well as death from COVID-19 [10,117]. COVID-19 has thus exacerbated the health inequality situation in the US [118,119] and beyond.

### 2.3. Climate Change and Health Inequity

Climate change is defined by the Intergovernmental Panel on Climate change (IPCC), the world’s leading scientific body on climate change, as naturally occurring or human induced changes in natural variability and climate disruption from anthropogenic forcing such as greenhouse gas emissions (GHGE) [120]. Anthropogenic climate change is a public health crisis that is a crucial current threat to global health and health equity [1,121,122]. While we are all at risk, the effects of climate change disproportionately impact the world’s most vulnerable populations, including populations of color in the US via increased risk for freshwater and coastal flooding, urban heat, and disasters such as hurricanes [123]. From 1980 alone, the US sustained 285 weather and climate disasters at a total cost exceeding USD 1.875 trillion (about USD 5,800 per person in the US) [124], with the year 2020 having record number of 22-billion-dollar weather and climate-related extreme weather events (EWE) that killed more than 262 people per reports that we know of [125]. Heat stress and EWE not only directly harm health but make many other conditions worse, such as dementia [126].

Globally, discussions on both climate justice and EJ have focused on differences across countries regarding responsibilities for causing climate change and the consequent responsibilities for mitigation and adaptation efforts [127]. High income countries have been the major parties responsible for GHGE while low – and middle-income countries (LMICs) suffer most from the resulting diseases and injuries, with the added burden of being called to participate in the global mitigation and adaption strategies, regardless of their minimal contributions [123,128]. This has led to conflicts between wealthy and poor nations of the global North and South [129]. Islam et al. (2017) focused on *“within countries inequalities”* and arrived at the same conclusion that poor individuals within all countries are also disproportionately affected by negative consequences of climate change adverse events [127].

Climate change disruptions promote EWE [122] including extreme hot and cold temperatures [130,131,132], droughts and desertification, thunderstorms, heavier precipitation, sea-level rise, and floods, ice melting and loss of ice sheets and snow-cover, tropical cyclones, worsening air quality, forest/wildfires, and dust storms [122,133]. EWE have serious public health consequences such as disruptions in water supply systems [134,135,136] and reduced access to drinking water, ocean acidification and disruptions in marine ecosystems [137,138,139,140,141,142,143], increased crop damaging pests leading to disruptions of agriculture and food supply systems and decreased food security [144], increased food-borne and water-borne diseases [145,146], increased vector-borne and zoonotic diseases [133,147,148,149,150,151,152,153,154,155], and other adverse deleterious effects on human health including increase in chronic diseases [156,157], such as CVD morbidity and mortality [158,159,160] and other loss of lives and livelihoods [122,133]. The health costs from climate-related premature deaths, need for prescription medications, lost productivity and wages, treatment in hospitals and clinics, home health care, among other downstream consequences are astounding, exceeding USD 800 billion per year in the US, and expected to increase if there is no immediate action [161]. Other negative economic consequences from EWE include irreversible damages such as loss of ecosystems and biodiversity, loss of indigenous cultural practices, loss of traditions and cultural identity, loss of land and property, coastal erosion and damages to buildings and other infrastructure [162,163], and disruptions in agricultural productivity and natural capital [122,133].

The negative effects of climate change disproportionately affect disadvantaged individuals living in poorer communities around the world, due to limited income resources and other adaptive capacities [1,122,127,164,165]. The United Nations Department of Economic and Social Affairs identified pathways that exacerbate inequalities in adverse effects of climate change or “climate hazards” in their 2017 working paper [127]. Existing social inequalities increase *exposure risk* for disadvantaged groups to climate hazards. The harms frequently combine and amplify risks such as that of flooding contributing to Per- and polyfluoroalkyl substances (PFAS) exposure and subsequent water contamination. In addition to the increased exposure level, social inequalities also increase disadvantaged groups’ *susceptibility* to climate hazards while decreasing their ability to *cope* with and *recover* from the damages suffered [127]. During climate hazards, disadvantaged individuals tend to disproportionately lose income and assets (i.e., physical, financial, human, and social) and suffer damages, which exacerbates inequalities and perpetuates a vicious cycle of climate risks [127].

This *“involuntary exposure”* to many societies has been described by scholars as *“possibly the largest health inequity of our time”* [1], and an opportunity for public health action that should not be missed [166]. Vulnerable populations including children and pregnant women, ethnic minorities, and indigenous communities [167], urban residents and outdoor workers, older individuals, and seniors as well as those with existing chronic conditions such as CVD or cancer and those with disabilities bear an increased risk for adverse health consequences of climate change [122,123,128]. In the US, individuals over the age of 65 are disproportionately impacted by heatwaves [168]. Older adults experienced over 102 million more days of heatwave exposure in 2019 compared with the 1986–2005 baseline [169]. Heat-related mortality for older persons, has almost doubled during the last two decades, and reached the record high of almost 19,000 deaths in 2018 [169]. A 2012 systematic review and meta-analysis of studies across different continents found that every 1 °C increment or decrease in hot or cold temperature periods was respectively associated with a 2–5% and 1–2% increase in all-cause mortality among the elderly [170]. Increases in summer temperatures have also been associated with increased mortality risk among older people with chronic diseases, particularly CVD [171]. Urban areas with electrical grid failures during times of extreme heat increasingly expose large populations to dangerously hot environments [172].

Another example is the disproportionate impact of fossil fuel air pollution among communities of color, including Black, Latinx [173], and Indigenous people [174]. In the most recent State of the Air Report, more than 40% of Americans (over 135 million people) live in places with unhealthy air, and people of color are three times more likely to live in these polluted spaces [175]. Fossil fuel air pollution is particularly prevalent in these communities with unequal burden for those living closest [176]. The pollution from natural gas flaring has been linked with up to a 50% increase in preterm birth in mothers living near oil and gas wells, most of them being Latina [177,178,179]. Even the location of natural gas pipelines is positively associated with social vulnerability, highlighting the link between health hazards and energy infrastructure [180]. Other negative health outcomes of natural gas flaring include increase pediatric asthma exacerbations leading to an increase in emergency department visits [181], and hospitalizations [182]. Food insecurity resulting from disrupted agriculture and food production systems [183,184] and a higher burden of a wide range of diseases attributable to climate disruptions [1] such as infectious diseases and cardiopulmonary diseases such as asthma and COPD disproportionately affect poor individuals in LMICs. Similarly, poorer communities within developed countries have increased exposure to adverse effects of EWE and increased susceptibility resulting damages [123]. For example, rising temperatures increase urban heat, which adversely impacts poor urban residents who have limited access to greenspaces that offer cooling effects [185,186].

### 2.4. Persistent Chronic Disease Burden and Inequity

Chronic diseases pose a persistent threat to global public health and health equity. According to the World Health Organization (WHO)’s Global Health Estimates, chronic diseases constituted 7 of the top 10 leading causes of global mortality in 2019 [187]. The top two on the 2019 WHO list are both CVD; ischemic heart disease (IHD) which is responsible for 16% of the world’s total deaths and stroke, which is responsible for 11% of total deaths respectively [187].

In the US, CVD remains the leading cause of death [188,189,190,191,192] followed by cancer [188,192,193]. Although initial reports had suggested that COVID-19 was the number one cause of death in 2020 [194], CVD and cancer are still the top two leading cause of mortality [195], responsible for 690,882 and 598,932 deaths respectively [192]. COVID-19 was the third leading cause of death, responsible for 345,323 deaths [192]. The American Heart Association (AHA) projections indicate that by 2030, 43.9% of the US adult population will have some form of CVD [190]. Similarly, the American Cancer Society (ACS) projects that by 2050, individuals 65 years and older will be particularly affected due to the doubling of new cancer cases among them because of their increase vulnerability [196]. Additionally, the close co-morbid linkages between CVD and cancer puts the patient of one of those diseases at an increased risk of having the other [197] due to shared risk factors [198], such as smoking, inadequate physical activity (PA), poor dietary habits, and obesity, which increase their co-occurrence in many patients [197] and disproportionately affect communities of color [199].

Additionally, CVD and cancer pose a significant burden to US healthcare costs [200,201]. They take a huge portion of the nation’s USD 3.8 trillion annual healthcare expenditure [200,201,202]. CVD costs USD 214 billion in annual cost to the US healthcare system and causes USD 138 billion in lost productivity on the job [203]. The cost of cancer care has also continuously risen, and the National Cancer Institute (NCI) had projected that the cost would reach USD 174 billion in 2020 [201,204]. In 2019, cancer patients spent nearly four-times higher per person (USD 16, 346) compared to those without cancer (USD 4,484) in mean healthcare expenditure [202].

Chronic diseases inequities have been exacerbated by structural issues such as systemic racism. Historical restrictive racial covenants and redlining, neighborhood disinvestments, and poor social and built neighborhood environments collectively contributed to a web of neighborhood disadvantages and created adverse social conditions that increased exposure and risk of disease for individuals living in those neighborhoods [205,206]. People of color are more likely to live in poorer neighborhoods [207]. Additionally, environmental injustice has also reinforced the purposeful siting of environmental polluting facilities in communities of color [208]. These social conditions increase the risks for CVD [209]. Consequently, minority populations receive the highest environmental burden that adversely impacts health, including both CVD [209,210,211,212,213] and cancer outcomes [214,215,216,217].

Both social and built neighborhood environments are essential determinants of health [218] and have direct implications in the entire cancer continuum [219] as well as CVD risk factors [210,220,221,222] including chronic stress [223,224,225,226] and obesogenic behaviors [227] such as poor food choices due to limited access to grocery stores “food deserts” or increased access to fast food retails “food swamps”, and inadequate PA [228] due to limited or poor-quality greenspaces [221,227,229] as well as neighborhood related poor safety concerns [230,231]. Yet, inadequate PA and obesity are the two main drivers behind the high prevalence of CVD in the US [232,233]. Inequities in the qualities of the social and built neighborhood environments [234] are linked to inequities in chronic diseases prevention efforts [235] such as healthy eating behaviors [236], PA and weight loss interventions [237,238] as well as other obesity prevention and reduction related interventions [239].

Additionally, being uninsured or underinsured [240] poses a barrier to health care services utilization including of CVD services [241] which exacerbates chronic disease risk and severity [207]. Consequently, limited access to insurance and neighborhood disadvantage intersects in predicting negative chronic diseases outcomes. Neighborhood disadvantage has been linked with increased risk for chronic diseases including CVD [242,243,244], increased comorbidity of CVD and other chronic diseases such as diabetes [243], and negative cancer outcomes [245] across the cancer care continuum including cancer risk [246,247], cancer prevention/screening [248], cancer diagnosis and incidence [249,250,251,252], cancer treatment [253], cancer survivorship [252,254,255], and cancer mortality [219,244,256,257].

Existing associations of chronic disease comorbidities and worse health outcomes as well as more complex clinical management [113] pre-disposed communities of color to poorer COVID-19 outcomes. COVID-19 and chronic disease comorbidity increased the risk of death for COVID-19 patients [258]. Communities of color including Hispanics, Indigenous populations, and African American populations had a disproportionate burden of COVID-19 mortality [81,117,259,260,261]. Subsequent studies linked structural racism, social vulnerabilities and long-standing systemic health and social inequities to the disproportionate burden of COVID-19 morbidity and mortality among racial and ethnic minorities [10,81,259].

Health professionals and the greater medical community needed to treat these diseases also have a long history of challenges that exacerbate racial injustices. The Flexner Report transformed medical education in the early 1900s in a way that disproportionately affected Black physicians [262]. Tuition and funding requests contributed to new financial hardships, and the closure of historically Black medical schools meant fewer opportunities for people of color [262]. The decisions had profound implications on a racially diverse physician workforce. In 2018, 12.8% of the US population was Black yet only 5.4% of physicians were Black; although there was no statistically significant difference in this percentage from 1940 to 2018 [263]. The Institute of Medicine, now National Academy of Medicine, highlighted disturbing data about worse quality of care provided in health care settings that directly contributed to longstanding racial and ethnic health disparities [264,265]. Engaging a diverse workforce is one component [265]. A study in California found that Black doctors could reduce the Black–White gap in cardiovascular mortality by as much as 19% [266]. Yet, other barriers persist, including gaps in salary. In 2018, the difference in median income between Black and White male physicians was USD 50,000 less for Black physicians [263,265]. In academic medicine, persistent lack of mentorship and funding exacerbates the systemic issues for Black physicians and threatens future success as defined by the traditional medical community for those who, against all odds remain in the medical profession [267].

### 2.5. Urgent, Intertwined Crises

Structural racism, COVID-19, and climate change all have adverse impacts on global health systems [268] and exacerbate endemic health inequities, including in chronic diseases [1,118]. A multifaceted and systems thinking approach is needed to build sustainable systems that will simultaneously enhance resilience and reduce health disparities and injustices. Such an approach may simultaneously address the aforementioned crises in the most efficient and effective manner collectively. The remainder of this paper will focus on the role of strategic investments in public urban greenspaces, broadly defined, as well as related activities to naturalize, activate, maintain, and leverage these spaces for ecosystem health, healthy building materials and workforce development opportunities, as an integral part of an antiracist strategy to build resiliency and reduce health disparities. The equitable distribution of outdoor green environments has the potential to reduce the health equity gaps between low-income and high-income neighborhoods that have kept people of color at a disproportionate disadvantage, even long after detrimental historical racists policies were implemented.

## 3. Urban Public Greenspace as a Critical Component of an Antiracist Strategy for Global Environmental Health Equity

Public greenspaces have numerous benefits for both physical and mental health [269,270,271,272], offering resiliency in the face of challenges posed by COVID-19, climate change, structural racism and combatting endemic chronic diseases. Building on a previous conceptual framework for health benefits of Climate change mitigation and adaptation [122], we propose a new conceptual framework for greenspace contribution to resilience and health equity in the US and beyond.

The next section will dive deeper into the key components of the proposed conceptual framework (Figure 1) with a focus on known health benefits of outdoor greenspaces particularly relevant to health inequities in COVID-19, climate change, and chronic diseases.

### 3.1. Greenspaces Enable Resilience for the COVID-19 Pandemic

Outdoor greenspaces facilitate social/physical distancing [273,274,275] while enhancing social connectedness [276]. Physical distancing is the primary and widely known strategy in reducing the risk for COVID-19 infection and transmission [274]. Outdoor spaces enable physical distancing and mitigate the spread of COVID-19 by reducing the risk of transmission [277]. Indeed, one scoping review conducted in late 2020 suggested that outdoor environments are associated with a reduced risk of SARS CoV-2 transmission [278]. Another review that investigated clusters of COVID-19 infections and their transmission settings linked very few infections to outdoor settings [279]. In addition to providing opportunity to physical distancing, greenspaces improve the social environment through urban community gardens [280]. Community gardens improve community connectedness and social capital [281,282], increase engagement in civic activities and individual’s connection to their cultural heritage and identity [283,284], and reduce crime rates while stabilizing neighborhoods [285]. Other studies have also reported positive associations between other greenspaces (not community gardens) and reduced violence, measured through reduced crimes rates [286,287,288] and reduced aggressive behaviors [271,289]. In the COVID-19 context, outdoor and greenspaces offer a safer place for low-risk social interactions [290,291,292], while facilitating physical distancing to reduce COVID-19 and other infectious disease transmissions. Greenspaces offer a potential resilience strategy for stakeholders to alleviate future COVID-19 home confinement while offering health benefits. Evidence suggests that if the park exists, it will be used. For example, three naturalized parks managed by the Urban Ecology Center (UEC) in Milwaukee, located in the mostly densely populated neighborhoods experienced double the usual visitation from March to September of 2020 during the COVID-19 pandemic. Additionally, weekend park attendance on 16–17 May 2020 was up over 44% compared to the previous year and continued to rise to 52% by 13–14 June 2020; and there was a sizeable increase in camping reservations [293].

Further, outdoor classrooms reduce the risk of COVID-19 infection transmission by promoting physical distancing while enabling children continued social interactions in the learning process [294,295]. These outdoor settings facilitate social/physical distancing, enhance social connectedness [276,296] and improve mental wellbeing [296]. Outdoor environments promote children’s social interactions [297,298] and enhance their health and wellbeing [297]. Similar beneficial relationships have been established in the adult population [299]. Outdoor greenspaces promote adult interactions and social cohesion [299] as well as social connectedness [291]. Social cohesion is associated with positive physical and mental health [300]. In the Health and Retirement Study of adults’ participants, 50 years and older, with no history of CVD (*n* = 5276), every standard deviation increase in perceived neighborhood social cohesion was associated with 22% reduced odds of myocardial infarction (MI) (OR = 0.78, 95% CI (0.63, 0.94)) even after adjusting for behavioral, biological, and psychosocial covariates [301].

Outdoor public greenspaces promote mental health and reduce stress. Greenspace has been associated with positive mental health outcomes including reduced likelihood of depressive symptoms [302,303], reduced stress [303,304,305], lower levels of anxiety symptoms [303], improved mood [306,307,308], and improved cognitive functioning [309]. Early on during the COVID-19 pandemic, mandatory home confinements had negative impact on physical and mental health in children [310] and in the general population [311,312,313,314,315]. Major stressors for children’s mental health [316] include fears of infection, frustration and boredom, lack of in-person contacts with their classmates, friends, and teachers [310]. In adults, COVID-19 increased the prevalence of depression, anxiety, and stress [311]. A scoping review conducted during the first 7 months of the pandemic, with a combined study population of 113,285 individuals, found a higher prevalence for all forms of depression at 20%, anxiety at 35%, and stress at 53% [311].

Empirical studies have linked residential greenspace to improved mental health during the COVID-19 pandemic. In a study with 556 residents from 15 residential communities in China, sociodemographic data were collected and the Kessler Psychological Distress Scale (K10 Scale) [317] was adopted to evaluate residents’ mental health status [318]. Authors found that green coverage ratio, satisfaction of the landscapes of greenspace, green view index outside the window, and green viewing duration of the residential greenspaces have positive effects on residents’ mental health status [318]. Greenspaces contribute to COVID-19 mental stress recovery process and ensure long-term resiliency for cities and municipalities regarding current and future health crises. Reductions in stress leads to reduced chronic disease risk and improved survival after chronic disease diagnoses [319,320,321]. Stress reduction has been linked with improved cancer survivorship [322,323,324,325,326,327,328,329] and improved CVD outcomes [330,331], including significantly reduced risk for mortality, heart attack, and stroke in coronary heart disease patients [332].

### 3.2. Greenspaces Improve Chronic Disease Outcomes and Can Reduce Health Inequities

In addition to promoting mental health, greenspaces also improve adults’ physical health outcomes via increased PA [271]. Studies have found positive associations between time spent in urban greenspace and moderate to vigorous PA (MVPA) [333,334,335,336]. For example, a study in Wisconsin investigated the salutogenic effect of trees on streets and sidewalks by assessing streets and overall greenery within 500 to 1250 m buffers zones at 424 addresses [337]. Authors consistently found that street tree cover was associated with increase in active transportation (AT) [337], a form of PA. After adjusting for all covariates, authors found that a 10% increase in street-level tree cover was associated with 19% to 41% increased odds of AT [337]. Studies have documented associations between greenspace, increased AT, and increased PA with positive outcomes in chronic diseases, through measures of reduced disease risks [292]. Examples include the reduction in risk for CVD [338,339], and CVD related mortality [340] including ischemic stroke [341], ischemic heart and cerebrovascular diseases [342], reduced risk of all-cause mortality [211,341,343], and reduced heart rate [307,308,344].

In regards to health equity, previous scholars have proposed the role of greenspace in reducing existing socioeconomic health inequalities [345,346]. Low income and urban neighborhoods with low- and poor-quality greenspaces [347,348,349] can benefit from the addition of quality greenspaces in the overall strategy. Benefits include improved air quality [321] and increased neighborhood safety. For example, urban trees have been associated with increased sense of neighborhood safety [350]. Investments in greenspaces in poor urban neighborhoods can reduce the existing endemic racial and ethnic disparities in greenspaces accessibility and use [349,351], reduce health disparities and improve health equity [305,352]. Further, research has indicated that trees can increase property values [353,354,355,356,357,358], a potential strategy in reducing the wealth equity gaps by increasing home values for those historically neglected neighborhoods [359].

Traditionally green-deprived built environments such as health care facilities and prisons would benefit from addition of greenspace [360]. These environments house large numbers of people and hence have exciting potential to leverage positive change. The US leads the world in number of people who are incarcerated [361]; and its health care is a leading industry employing millions of people [362]. In health care facilities, greenspaces offer a calming environment and comfort for patients and employees [363,364,365]. Health care providers may also suffer from negative mental health effects, particularly during times of increased stress to health systems such as during climate-related EWE [366] or the COVID-19 pandemic [315]. Community gardens affiliated with academic health centers and hospitals have been linked with reduced rates of obesity in communities [367]. In prisons, one study found that individuals who were experiencing incarceration made fewer sick calls when they had something green to look at than those who did not [368]. Watching nature videos was associated with a 26% reduction in violent behaviors [369]. Recidivism rates in the US are substantial and green prison programs in California and New York, among other programs that teach skills and lead to employment, have shown positive effects for participants and for reducing recidivism rates [370,371]. Additionally, in the general population, greenspaces have been linked with reduced neighborhood violence and crime [372].

Green schoolyards have also been associated with children’s positive physical health outcomes including reduced sedentary behaviors [373,374], improved wellbeing and cognitive performance [375], reduced physiological stress [376] and increased levels of PA [298,373,377,378] through different measures such as increase in time spent in MVPA, reduction of time spent in sedentary PA, increase in number of children observed in MVPA and reduced number of children observed in sedentary activities [298]. Children’s PA is a well-established mechanism in preventing numerous adulthood diseases such as CVD [229,379,380], type 2 diabetes [379], overweight and obesity [229,380,381,382], and psychological disorders [383], among others.

In addition to physical health benefits, children’s exposure to greenspace has also been associated with improved social wellbeing, and positive mental health outcomes such as reduced depression [384,385], reduced anxiety [385], improvements in self-esteem [385], and enhancement in psychological wellbeing [376]. Greenspace has also been linked with numerous beneficial effects in children’s socioemotional health [298] including increase in positive behaviors and social interactions, reduction in verbal and physical conflict rates and improvement in attention restoration, particularly observed after exposure to greened schoolyards [298]. School based greenspaces offer an opportunity for fostering health equity for children from low-income families who are increasingly growing up in urban areas with limited access to nature [298,386]. Additionally, children’s early exposure to greenspaces through outdoor environmental education potentially promotes children’s positive attitude towards outdoor play [387]. Outdoor environmental education that leverages local greenspaces fosters children’s environmental stewardship and grooms them into future environment friendly citizens who will be ready to push forward good and equitable policies for climate change mitigation and adaptation to protect generations to come.

### 3.3. Greenspaces Are Essential Components of Climate Change Mitigation and Adaptation

Greenspaces contribute to climate change mitigation and adaption strategies [122]. Greenspaces increase systems’ resilience to climate change impacts [388]. Greenspaces also cool down cities [185,186] and reduce urban heat island effects by shading building surfaces and deflecting radiation from the sun [389]. Greenspaces also reduce air pollution and respiratory system diseases by reducing indoor and outdoor traffic related air pollutants (TRAP) levels [390]. Greenspaces, in particular trees, sequester carbon by absorbing it for photosynthesis reduce atmospheric CO_2_, and mitigate flooding by absorbing excess rainwater [185,391]. Green roofs reduce energy consumption and improve stormwater management and water quality [392,393].

Quality urban greenspaces offer an opportunity to respond to the challenge of food insecurity and improve nutritional health through community gardens and edible forests [394]. Urban community gardens increase urban biodiversity and access to ecologically sustainable and healthier foods including fiber rich fruits and vegetables [394]. Participating in community gardening has been associated with increased consumptions of fruits and vegetables among urban residents [395,396]. Additionally, school gardening has been associated with improved knowledge and attitudes toward vegetables and fruit among children [397]. Increasing community gardens has a positive impact on nutrient intake among adults and children, which has been associated with numerous health outcomes including positive CVD and cancer outcomes, among others [398,399]. Community gardens have also been associated with improved physical health among vulnerable groups, including refugees [400]. Well designed and managed greenspaces can advance health equity by increasing access to healthier foods choices in low-income urban food deserts and food swamps.

## 4. The Way Forward: Harnessing a Moment of Political Will to Make Strategic Investments in Equitable Urban Greenspaces and Trees

With multiple urgent and intertwined crises, there is a need for immediate action. We propose that a strategic investment in equitable urban greenspaces be an important part of the solution to the challenges we face now and represents an opportunity to address research gaps to guide future evidence-based public health practices [401]. At the time of this writing, the US has recently inaugurated a new political administration (Biden–Harris), providing a potential opportunity and political will to move such an agenda forward. Combatting COVID-19, structural racism, and climate change are the top three among key priorities of the Biden–Harris administration [402].

On 19 December 2020, the then-US Vice President-elect Kamala Harris reinforced how climate change is an existential threat to all of us, disproportionately affecting poor communities and communities of color who are at increased risk for polluted water, polluted air, and a failing infrastructure [403]. She then reiterated that everyone has the right to drink clean water and breathe clean air and briefly talked about their administration Climate change ambition of building the path to achieve net-zero emissions, by no later than 2050, starting by recommitting the US to the Paris Agreement on climate change. In addition to rejoining the Paris accord, the new administration blueprint includes targeting all economic sectors in their “building back better plan,” including infrastructure, auto industry, transit, power sector, building, housing, innovation, agriculture, conservation, and EJ in building safer, healthier, thriving, more resilient, and sustainable communities [404]. She quoted the Holy Father Pope Francis in his 2015 Encyclical when he said: *“Humanity still has the ability to work together in building our common home”* [405]. The vice-president elect concluded her remarks by reinforcing their plan in putting a concerted effort starting 20 January 2021, in heeding the Holy Father’s words by working together nationally in the US and involving other global leaders in building and protecting our common home for our generation and generations to come [403]. Similarly, in his inaugural speech on 20 January 2021, the 46th US President, Joe Biden, reiterated his commitment to fighting climate change, COVID-19, and structural racism [406]. Those Climate commitments were affirmed when the US officially rejoined the Paris Agreement [407] and a subsequently calling a Global Leaders Summit on Climate Change to pave the way for the 26th Conferences of Parities (COP26) in Glasgow, UK [408].

Concurrently, there has been a surge of environmental bills from Congress, including the Tree Act (S.4038, HR 8291), the Climate Stewardship Act of 2019 Title 2: REFORESTATION TRUST FUND. (HR4269), Preventing HEAT Illness and Deaths Act of 2020 (S4280), and the 21st Century Civilian Conservation Corps Act (S4434), with funds to plant over 100 million trees across America. This political will is likely to result in increased funding to local governments to address specific local threats to climate change. There is an urgent need of an equitable and fair distribution of those investments based with a focus on populations who are at increased risk of climate change vulnerabilities.

The IPCC in 2018 pointed out that in addition to emission reduction strategy, which has been the main climate strategy for decades, there is need to put greater emphasis on carbon control through natural systems [409,410,411]. Therefore, in urban context, urban reforestation provides the greatest opportunity to address climate change through carbon sequestration and has additional benefits for storm water management and air quality improvement [412]. For example, the City of Boulder, Colorado, along with community partners, developed a plan for urban forestry with tree canopies to adapt to the effects of climate change [413].

Managing carbon through natural landscapes in urban areas, which also provides ecosystem services to promote health equity, is now on the policy front and many initiatives have offered proposals to the new Biden–Harris administration [414]. The Urban Sustainability Directors Network (USDN) and Carbon Neutral Cities Alliance (CNCA) are among networks of cities that are launching initiatives regarding carbon management in cities through natural landscapes. Similarly, urban Drawdown Initiative, USDN, Trust for Public Land, American Forests, and Davey Trees all have initiatives around policy proposals to expand urban forests for both addressing the climate crisis and as an opportunity for green economic recovery by bringing money into communities to foster equitable economic development in cities.

The already expressed political will is an opportunity for public health professionals, atmospheric and earth scientists, and health care providers to meet the present challenges and act in a united front to solve unique intertwined global health crises. Greening, including planting and maintaining urban trees, must be an essential part of an antiracist strategy for global environmental health equity. These efforts must be equity focused and cognizant of historical injustices. Otherwise, there is a risk of magnifying inequity.

## 5. Key Strategies and Considerations: Examples from Milwaukee, Wisconsin

### 5.1. Recognize and Carefully Consider Distinct Types of Greenspaces and Their Various Strengths and Drawbacks before Investing

Here, we define public urban greenspace broadly, to include urban trees, forests, parks, natural areas, community gardens, and green infrastructure that are accessible to communities in urban areas. These types of greenspaces are not equal in definition, nor public health impact, maintenance, or relevance to different communities. Without qualification, referring to greenspace may simply stir up images of freshly mown lawns to some but intact ecosystems to others. Meaningful change may include some of each, but the type of greenspace selected for a particular endeavor will depend on the larger purpose of that endeavor, and communication is key.

Greenspace design may seek to maximize efficiency and benefits for climate change mitigation and adaptation. For example, not all trees are equal in terms of carbon sequestration or water absorption [415,416]. Some tree species sequester more carbon than others [416] while other species are more efficient in stormwater capture [417] (deciduous vs. coniferous, big leaves vs. small leaves) and potential cooling (deciduous broad leaf = more shade = more cooling [418,419]). For example, trees species with higher tolerances to urban stresses and higher wood densities have been found to have the highest carbon sequestering capacity [420]. Greenspaces may also seek to support local community needs, including for fresh fruits and vegetables via community gardens or for sports and recreation areas, including pocket parks and sports fields. Greenspaces may also be targeted to reduce the urban heat island effect or mitigate air pollution along busy roadways.

Additionally, some tree species can be the source of allergenic pollen that can potentially cause asthma, allergic disease, and respiratory infections [421,422]. It is therefore very important in carefully selecting tree species to reduce the risk of allergies and associated diseases. Urban planners need to ensure that climate change and health benefits from trees are not a sacrifice for potential downsides such as pollen allergies. Allergenic tree species such as the genus Platanus, or London plane tree should be avoided to reduce adverse effects of allergenic pollen [422].

Greenspace design may seek to maximize ecological integrity. Nature is critically important in ensuring biodiversity. Yet, there is a significant worldwide species decline in animals, birds, plants, and insects [423,424,425,426,427,428]. When urban land is naturalized; the benefits go beyond the increase in human use of the space. The wildlife usage increases significantly as well. In Milwaukee for example, in one of the parks managed by the UEC, UEC and other stakeholders began seasonal targeted bird monitoring in 2007 and regular weekly monitoring in 2011. The cumulative species bird count since 2011 has grown by about 50% and the average species seen per year has increased by 18%. This increase in species diversity corresponds to the increased efforts of land stewardship and the enhancement of native plant community types in the greenspace that are important to birds and were not present prior to UEC land management. New species of dragon flies, bees and other insects, amphibians, reptiles, and mammals have been regularly identified in these parks with this management. Fish populations have exploded in the restored rivers, offering both food and recreation to residents and other species like the beaver, not seen in Milwaukee for over a century, are now back as a regular resident of the city. Hyper local naturalization of urban greenspace can play a significant role in species preservation.

### 5.2. Focus on Equity, Prepare for, and Prevent Green Gentrification

History has taught us how investments and disinvestments drive inequities. Stakeholders must be careful in managing new greenspace investments to ensure that they emphasize equity in outcome, rather than equality in distribution [429]. If not targeted and deliberate, there is a risk of magnifying existing inequities. Recognition is the first necessary step toward this goal, and it is a determining factor in contemporary justice theories; as maldistribution in many cases is the result of misrecognition and consequent disempowerment [430].

Additionally, green gentrification is sometimes an unintended consequence of greening because greenspaces increase property values while are being created [431,432,433,434]. Concerns about green gentrification need to be considered and addressed [435]. Proactive policies or systems should be established to ensure that the people who live near newly emerging greenspaces can continue to live in their homes and benefit from the greenspaces that are being created to ensure equitable access. Some of the potential pro actions include aggressive construction of low-middle-income housing, reducing property taxes to protect long-time residents, ensuring protection for senior homeowners, and creating stabilization vouchers [436].

Equity can be accomplished through community-engaged interventions to ensure representation. Another important consideration is objective evaluation of the equity outcomes of urban greening efforts. PROGRESS-plus is an example of such evaluation tools [437].

### 5.3. Consider Plantable Spaces and Ensure Connectivity for Greenspaces Safer Use

Greenspaces in urban areas are so important but underserved urban areas are often the most densely populated areas and therefore land for potential greenspaces is limited. To achieve the goal of integrating more greenspaces in these areas, urban planners may have to think more creatively about how and where to integrate those greenspaces. Greenspace connectivity is also important especially in underserved urban areas. If people cannot safely get to the greenspace, or they do not know where the greenspace is, or they can only arrive by car, they will not use it [438]. Connectivity provides opportunities for cities to prioritize building complete streets or safe streets infrastructure that provide safe routes that residents can take to get to new greenspaces being created [439]. Reckless driving and traffic are problems and if people do not feel like there is a safe route to get to a greenspace, the safest thing for them to do is not to use it [440]. The stress of safely getting to and through the greenspace should not outweigh the benefit of interacting with the greenspace. If the goal is to create the greenspaces that people want to use, the ease of getting to and away from the space should be considered, including creating safer routes with clear signage.

### 5.4. Activate Greenspaces for Safety and Maximal Health Benefit

Activating a greenspace with positive programming, resources, and engagement is key to the success of the space. Including design features to encourage PA is one of the most common strategies to activate a space that will also increase visibility and safety; in accordance with the theory of crime prevention through environmental design [441]. Further, innovative approaches to establish facilities with programming and equipment to support activated greenspaces have shown promise. A well-designed greenspace can encourage active use, but to really see results it is best for a greenspace to have an anchor tenant or partner to facilitate activity. In Milwaukee for example, there are numerous groups that have emerged to fill the niche for greenspace activation, including beer gardens, nature centers, nonprofits, churches, and neighborhood organizations. In some places this activation work is accomplished through the public sector in parks departments or city parks districts, while in others it is done by local nonprofits. When putting greenspace plans together, the activation component needs to be prioritized in equal measure to design.

The UEC, in partnership with other stakeholders, has successfully converted abandoned, environmentally degraded former industrial land into high quality naturalized park land accessible to neighborhoods that did not previously have access to the natural assets of their community. This newly activated land in an otherwise highly developed neighborhood attracted investments, businesses, field trips from schools, and tremendous park use from a broad spectrum of community members. With creative urban planning, quality greenspaces can be created in densely populated areas and are good long-term investments for those neighborhoods.

### 5.5. Leverage the Greenspace for Environmental Education to Build a Culture of Environmental Stewardship

Greenspaces are valuable assets that can be leveraged to build a culture of environmental stewardship through environmental education programming for school children and adults, alike [442,443,444,445,446]. Public parks and schoolyards themselves can be leveraged for this purpose, teaching students about environmental science, environmental justice and focusing on hands-on, experiential learning to bring the messages home. Research shows that early life exposure to nature and mentorship are important influences in building an environmental ethic [442,447,448,449].

Schoolyard transformations can offer direct engagement with nature in a public education setting via outdoor classrooms. Many Milwaukee-area schoolyards are covered in crumbling asphalt and offer little if any greenspace. These schoolyards were originally paved to reduce lawn maintenance costs in a low-income school district, but decades of inertia have left behind deteriorating infrastructure that is costly to repair, damaging to the urban environment, and unsupportive of children’s health at school. An increasing body of research indicates that students’ access to green schoolyards can result in better academic outcomes [450,451,452,453], increased engagement and enthusiasm [454], improved socioemotional wellbeing [298,386,455], improved physical health [298,386], and meaningful STEAM (science, technology, engineering, arts, and math) curricular connections [453]. Green schoolyards also serve as local assets in climate change mitigation and adaptation strategies [122] including in stormwater management and urban biodiversity [456].

### 5.6. Maintenance Planning, and How It Will Be Funded, Should Be Discussed from the Start

Concrete plans for the long-term maintenance of trees should be developed and implemented. Professional tree care is necessary so that trees can achieve maturity and maximize potential benefits. As the cost of professional tree maintenance can be substantial, it must be accounted for so that the cost burden is not shifted to those who can least afford it. Options include having the municipality/nonprofit take full responsibility for the trees they plant or the municipality/nonprofit offering cost-sharing for maintenance and removals whenever necessary. An appropriate tree ordinance should be developed that clarifies what the municipality is responsible for and what residents are responsible for regarding both public and private trees. This document should be easy to understand and enforceable by municipal staff.

Many communities lack the skills, resources, and technical expertise to create and sustain a healthy urban forest. There is a need for arborists to partner with those communities in need so that individual trees are maintained. There is also a need of urban foresters to maintain the urban forest as a whole or develop this expertise in-house. Maintaining the urban forest is a crucial part of ensuring its longevity and maximizing its benefits. Tools such as tree inventories should be employed. Tree inventories allow for the planning and tracking of species diversity and age diversity, two components of a healthy urban forest [457]. Tree inventories and other data systems are also essential for developing maintenance budgets and schedules, which depend on factors such as the number of trees, species, size, and condition.

### 5.7. Consider, Link to, or Embed Workforce Development Opportunities

Studies have shown that investing in greenspaces creates more employment opportunities than other type of investments. For example, a study at the Political Economy Research Institute at the University of Massachusetts found that every USD 1 million invested in forest management and restoration creates nearly 40 full-time employment positions, two to three times more jobs compared to other sectors’ investments [458]. Additionally, it should be an opportunity for training future workers to design, install, and maintain this infrastructure, through internships and other local future workforce development opportunities. Opportunities may be particularly valuable in low-income urban neighborhoods where there is not only a need for trees and parks, but for jobs, education, and economic opportunities overall.

People living in disinvested urban neighborhoods may not realize that there are potential jobs within arboriculture and urban forestry, among other greenspace associated professions. There is a need to instill the next generation with an understanding of environmental issues and place these issues in the historical context that created them while creating green jobs [459]. Exciting opportunities exist to train new workers in the fast-growing arboriculture and urban forestry industries that need more skilled workers and are experiencing a labor shortage with their traditional employee pipelines. Sector partnership programs have shown to be successful in connecting employers and low-income urban workers not traditionally engaged in the sector. Sector partnerships are centered around non-profit agencies who recruit candidates into work force training programs that provide skilled training and experiences in the industry that make candidates more attractive to employers. Many of these programs are done within low-income neighborhoods. Often funded through grants, these programs can act as catalysts for job creation, diversity, and awareness of the value of the urban forest. Workforce opportunities also exist for environmental educators, scientists, and environmental health professionals. For example, in Milwaukee, UEC emerged from the need for urban greenspaces 20 years ago and now employs annually over 80 people a year—educators, foresters, land stewards, and researchers—while supporting activity for numerous volunteers each year. 

### 5.8. Plan for Waste and Harness the Promise of Urban Wood

Urban wood refers to wood from trees removed for reasons other than harvest and comes from forests in backyards [460]. Examples of this wood primarily include logs from municipal or private landscape tree removals, trees damaged by EWE such as storms, trees damaged during pest outbreaks or those cut down during construction [461]. Urban wood as a large source of biomass is underutilized [462]. Most urban wood used to be disposed of as waste and therefore contributed to municipal waste instead of being used as a source of biomass or for many other purposes, including in the construction and interior design industries. Research estimates the potential annual value from US urban tree wood waste between USD 89 and USD 786 million [463]. There are many incentives for utilization of urban wood including disposal fees, environmental sustainability, and creation of revenue sources [463,464]. However, transitioning from disposal as municipal waste to utilization of urban wood affects a wide range of stakeholders and local strategies are essential in building required partnerships to repurpose urban wood based on specific local government (i.e., counties, cities, and municipalities) contexts [465].

When trees must be removed from our communities, many of their benefits as living trees carry on through as urban wood products [460]. There is increasing recognition of the value of natural building materials with biophilic design approaches [466]. Urban wood can be recycled instead of being wasted in landfills [460]. Urban wood can be used in building design, furniture design, floors, windows, cabinets, interiors finishes, and doors to improve physical, mental, and social emotional wellbeing [461,466]. Utilization of recycled urban wood for biophilic design can also potentially withhold a considerable amount of CO_2_ from the atmosphere [467]. Approximately, a white oak dining room table with ten chairs for example sequesters the equivalent of 730 pounds of CO_2_ [467].

As trees continue to be removed from urban and community forests due to pests and diseases, old age, or other factors such as economic development, there is opportunity to utilize these materials to their highest and best use to maximize economic, environmental, and societal benefits for urban communities, their residents, and their visitors. Changing attitudes and practices to maximize utilization of urban forest products would develop a supply of raw material that could feed both established and newly created markets for urban wood [468]. Education, awareness, and market development for urban sourced materials among architects, interior designers, and consumers, among others, is essential to the viability of urban wood utilization [469]. Finally, the urban wood industry can offer new opportunities for workforce development.

### 5.9. Engage Communities, Local Partners, and Key Stakeholders Early and Often

Multiple partners, local leaders, and community members must work together to maintain and grow healthy urban greenspaces. Engaging residents and neighborhood leaders in urban greening projects is of paramount importance [470]. Multiple case studies have shown that when local government or an outside nonprofit organization plants trees, redevelops a vacant urban lot, or installs a local park without neighborhood-level engagement, it can be met with apprehension and a lack of support [470,471]. Success should be measured not only by the number of trees planted, but by the level of community engagement [472,473]. Because community engagement can make or break the success of a project, communities should be engaged early and often, and for several purposes.

#### 5.9.1. Engage to Fund and Sustain Investment

To build successful investments in healthy greenspaces, multiple stakeholders need to work together to ensure efficient use of resources. In Milwaukee for example, the Milwaukee Metropolitan Sewerage District (MMSD) is funding the nonprofit organization Reflo to improve community resilience by transforming hard concrete schoolyards into greener and healthier yards through installation of cisterns, bioswales, outdoor classrooms, gardens and trees that will absorb storm water, reduce flood risk, and provide community benefits including by promoting social interaction, PA, and mental health [474]. Similarly, the Menomonee Valley Partners (MVP) has partnered with the UEC in the transformation of 24-Acre Brownfield into an active urban park, state trail, and outdoor science classroom [475]. They also turned a vacant bar into a community center and reconnected communities separated for more than a generation [475]. Such partnerships involved pulling funds together for great community investment and create a situation where everyone benefits.

#### 5.9.2. Engage to Educate

Educating the public about the benefits of greenspace is a vital part of community engagement in urban greening. Local leaders may need to be convinced that greenspaces are worth investing in. Additionally, because most of the land in what consists of an urban forest is privately owned [476], it is important to involve community members in planting and caring for trees on private property as well as on public property. Public education on tree planting/care and the benefits of trees is a crucial part of community involvement, especially for efforts focused on private trees.

#### 5.9.3. Engage on Design and to Find a “Fit”

Including residents as a stakeholder group for local greenspace design is essential. This ensures that greenspaces are designed with community members in mind. Otherwise, failing to meet the community’s needs might lead to underutilization and loss of intended benefits. Groundwork Milwaukee has offered some promising practices. All more than 80 active community gardens in their network were initiated by groups of residents and/or community groups. Before issuing any lease for a garden and building a garden, the Groundwork team ensures that there is sufficient community buy-in to sustain the space. The process involves designating a garden leader and assigning them the task of canvassing in their neighborhood. Experience has shown that gardens tend to have longer lifespans when more people are involved in their conception.

#### 5.9.4. Engage around Ownership, Use and Maintenance

Greenspaces cannot just be built; they must be actively sustained. Urban tree survival is essential for sustainable urban forest and ecosystem management [477]. Initial planning and consideration through selection and planting and long-term monitoring are essential for long term maintenance of both trees and their benefits [478]. Long-term maintenance requires local ownership. Previous work has shown that local stewardship is essential in both planting and maintenance practices [479]. Therefore, local leaders must put together a team of multiple invested stakeholders to ensure continuous maintenance of those greenspaces and safety for their use.

Innovative strategies to monetarily value the community effort invested in greenspace maintenance should be considered. Unfortunately, there is often no funding for maintenance of community led gardens. All garden leaders involved in maintaining Groundwork gardens in Milwaukee are volunteers, which can lead to burnout, causing gardens to turn inactive. A municipal program to financially support community garden leaders, as they care for public land (i.e., city owned vacant lots) may avoid later costs associated with cleaning up and repurposing a lot that contains a failed or inactive garden.

Community members must also play a vital role in ensuring greenspace quality. Quality matters, especially in underserved communities where residents are more used to their greenspaces being a grass lot versus a vibrant parkland. With a local example in Milwaukee, WI at Pulaski Park, residents wanted a futsal court rather than a basketball court; they wanted a sledding hill rather than having the entire slope forested; they wanted a jungle gym, they wanted trails, they wanted art and community gardens. Had the residents not been at the table when design plans were made, Pulaski Park would have had basketball courts (which they would not use), the slope would entirely be covered by trees (and they would have been disappointed and would not have used it in the winter), and the jungle gym and trail might have looked different than hoped.

Community based programming in greenspaces solidify them as community assets. For example, a pilot *“Community Garden Health Hub”* program in Milwaukee [480], is the most thorough investment in garden programming at Groundwork and will provide valuable insight into the ways in which programming can further transform greenspaces into valuable community assets. Investments will be made in engaged gardens with specific outcome measures in place to track the impact in detail. Conceptually, the model is very promising, and the team is expecting positive community and health outcomes.

## 6. Conclusions

Multiple threats from COVID-19, climate change and structural racism, in the context of endemic chronic disease disparities, contribute negatively to individuals’ physical and mental health as well as their overall wellbeing. These threats require a collective equity focused strategic response to ensure equal and equitable access to resources across all communities. This includes the often-overlooked growth, naturalization, and activation of public urban outdoor greenspaces. Outdoor green environments enable physical distancing and can serve as long-term strategic investments in reducing the risk of COVID-19 and other airborne viral infections while increasing resilience to global climate change. Additionally, greenspaces offer other benefits for both human and environmental health including enhancing biodiversity, improving mental and physical health, and preventing metabolic and chronic diseases such as cancer, diabetes, and cardiovascular diseases, among others.

The investments in quality urban parks and green infrastructure, including urban trees should target low-income urban neighborhoods with limited access to greenspaces, to bridge the existing endemic gap in equitable access to resources as well as the resulting inequities in health outcomes. Otherwise, there is a risk of magnifying existing racial and ethnic health disparities. Additionally, there is a need to invest in urban schools greening and environmental education to activate greenspace and increase exposure to nature for urban children to improve their physical health, mental health, and social wellbeing. These new investments will not only improve individual health and wellbeing but can create jobs, attract investments, and offer community resiliency in the face of the challenges posed by COVID-19, climate change and structural racism. They will aid the US in preparing for new challenges that will come with EWE, displacement and mobility of people, and other threats that have yet to be recognized.

It is important to be warry of lessons learned from the long history of the Federal government in marginalizing and discriminating against communities of color. What we have witnessed during the pandemic and the rise of social liberation movements such as Black Lives Matter and Red Deal [481] signify the importance of dismantling structural discrimination as we strategize for recovery. Workers labeled essential but deemed expendable by policy makers, incarcerated populations left without resources, and indigenous communities left behind in COVID-19 response [482] are all echoes of structural racism in housing, education, urban spaces, occupation, access to health care and food systems. We must learn from our mistakes and move forward deliberately to strategically deploy our greatest salutogenic assets across American neighborhoods with a focus on equity in effect, rather than equality in distribution. We must invest wisely, with communities at the table, in the right greenspaces in the right locations, infrastructure that gives the space purpose, maintenance and programming to ensure long-term health promotion, biodiversity and the urban ecosystems, with an eye to the future including workforce development, environmental education and the value of urban wood, and we must invest in the communities, and the community leaders, who care for and use these spaces. The new proposed conceptual framework should guide researchers who are interested in greenspace and health with an equity focused lens as the US envisions a healthier nation without health disparities.

## Figures and Tables

**Figure 1 ijerph-18-08420-f001:**
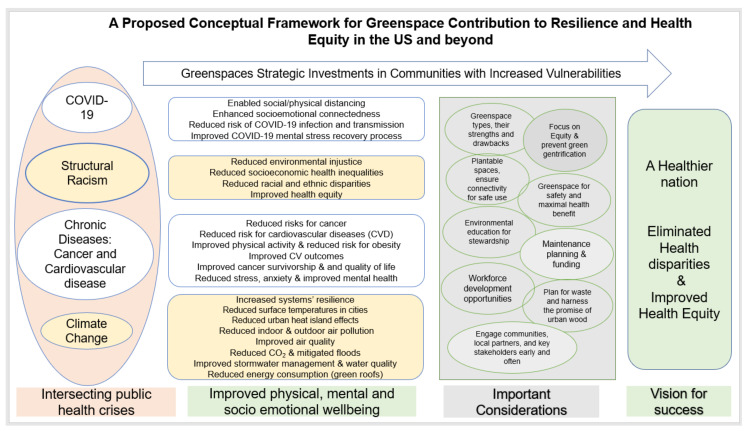
A proposed conceptual framework for greenspace contribution to resilience and health equity in the US and beyond.

## Data Availability

Not applicable.

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
