# Peer review of "Resilience and Equity in a Time of Crises: Investing in Public Urban Greenspace Is Now More Essential Than Ever in the US and Beyond"

_ijerph, 2021, doi:10.3390/ijerph18168420_

Round 1

Reviewer 1 Report

I really enjoyed this paper. It synthesizes multi-disciplinary information and uses multiple lines of critical research evidence to promote green spaces for the future. The researchers present a deep and compelling body of evidence for this position paper, and the English language in this is excellent. I couldn't see any obvious grammar or spelling errors, which is great for a reader. I commend their efforts on this paper.

Comments / suggestions:

It's incredibly America-centric. Almost all of the evidence presented in this review is concentrated on one part of the world, to the point where it might be worth re-titling the paper around this or at the very least noting this focus in the Abstract. It makes me wonder if the points made in this paper are (or aren't?) more broadly applicable to other nations which have very different legal, social, and environmental systems and histories. I think it's absolutely fine to focus just on America, but specify this up front for readers.

The formatting is very dense for a review paper of this kind. The crux of the paper seems to lie within "3. Urban public green space as a critical component of an anti-racist strategy for global environmental health equity" and beyond, but it's only bold font that organizes each point. May I suggest sub-headings rather than bold? This will really help your logic flow better. It's all there for readers, just not organized as clearly as it could be.

Is there a way you can create a visual that connects your mechanisms together, so that this is more accessible for the public? This paper proposes interactions between social and spatial components of our community. I think it would add value if you can show this connectivity.

Let's play devils advocate: Are there any downsides to green spaces worth discussing? Surely yet? The public can be wary of green spaces because of perceptions of pest animals, costs, maintenance efforts, etc. This paper is so strongly focused on the benefits, that I can't see much mention of the potential downsides. I think the authors could consider adding this dimension for balance. It won't undermine your points.

Those are my points of feedback, but again, I really enjoyed this paper and I'm certain this will generate a lot of interest and excitement when it's published.

Author Response

Thank you for the thoughtful and careful review of our manuscript. It has improved our manuscript a lot. 

Reviewer 2 Report

It is very clear that the work described in the paper is very extensively researched and presented in a way that developed some strong arguments. The paper would benefit, in terms of clarity for the reader, from having a section devoted to the case study "Milwaukee" as this does not come across very clearly in the paper.

The English language and style are fine.

With regard to the recommendation, I would like to see some minor revisions to reflect the comment regarding the case study above, but the substance of the paper fine.

Author Response

(The authors gave the same response as above.)
